# Surgical Strategies for Recurrent Hepatocellular Carcinoma after Resection: A Review of Current Evidence

**DOI:** 10.3390/cancers15020508

**Published:** 2023-01-13

**Authors:** Flavio Milana, Michela Anna Polidoro, Simone Famularo, Ana Lleo, Renzo Boldorini, Matteo Donadon, Guido Torzilli

**Affiliations:** 1Department of Biomedical Sciences, Humanitas University, 20072 Pieve Emanuele, MI, Italy; 2Department of Hepatobiliary and General Surgery, IRCCS Humanitas Research Hospital, 20089 Rozzano, MI, Italy; 3Hepatobiliary Immunopathology Laboratory, IRCCS Humanitas Research Hospital, 20089 Rozzano, MI, Italy; 4Department of Internal Medicine, IRCCS Humanitas Research Hospital, 20089 Rozzano, MI, Italy; 5Department of Health Sciences, Università del Piemonte Orientale, 28100 Novara, NO, Italy; 6Department of Pathology, University Maggiore Hospital, 28100 Novara, NO, Italy; 7Department of General Surgery, University Maggiore Hospital, 28100 Novara, NO, Italy

**Keywords:** hepatocellular carcinoma, recurrent hepatocellular carcinoma, liver resection, liver transplant, salvage liver transplant, surgical management

## Abstract

**Simple Summary:**

Among primary liver cancers, hepatocellular carcinoma (HCC) is the most common. Surgical resection and liver transplantation both represent potentially curative treatments not only in the case of the first occurrence, but also in those cases of disease recurrence if a proper selection of patients is performed ahead. Incidentally, the type and the time of relapse carry important weight on patient prognosis and overall survival. For these cases, proper management has still not been exactly defined. If precise indications for HCC first occurrence are quite clear, very few clear indications have been reported in those cases of relapse. The authors, after an extensive review of the published literature, aimed to summarize the modalities and the histopathological patterns of HCC recurrence, their prognostic value, and the main surgical strategies to deal with HCC relapse. At this point, either for redo hepatectomy or salvage liver transplantation, the pros and the cons have been detailed with the aim of characterizing the most suitable patients for receiving one or another. Some studies where such treatments were compared have been reported as well.

**Abstract:**

Hepatocellular carcinoma (HCC) is the most common primary liver cancer, and both liver resection and liver transplantation are considered potentially curative options. However, high recurrence rates affect the prognosis depending both on the primary HCC pathology characteristics or on the type and time of the relapse. While great attention has been usually posted on treatment algorithms for the first HCC, treatment algorithms for recurrent HCC (rHCC) are lacking. In these cases, surgery still represents a curative option with both redo hepatectomy and/or salvage liver transplantation, which are considered valid treatments in selected patients. In the current era of personalised medicine with promises of new systemic-targeted immuno-chemotherapies, we wished to perform a narrative review of the literature on the role of surgical strategies for rHCC.

## 1. Introduction

Hepatocellular carcinoma (HCC) is the most common primary liver tumour and the fifth leading cause of cancer-related death worldwide [1]. Most of the time, HCC develops on a liver cirrhosis substrate [2], with hepatitis B virus representing the most common risk factor in about 50% of cases [3]. The risk attributable to hepatitis C virus infection has been decreasing due to new antiviral drugs [3,4], while metabolic syndrome leading to fatty-liver disease and non-alcoholic steatohepatitis is becoming a remarkable growing cause, particularly in Western countries [5].

Primary liver resection (PLR) and primary liver transplant (PLT) are considered potentially curative options [6,7]. However, recurrent HCC (rHCC) after surgical treatment represents a major challenge with a median 5-year survival after recurrence (SAR) of about 35% [8].

To date, numerous articles have been published focusing on recurrence risk factors, showing many different independent variables [9,10], such as those related to tumour pathology (i.e., micro- or macrovascular invasion, presence of satellitosis, differentiation grade) and those related to the underlying liver substrates (i.e., grade of fibrosis, hepatitis viral infections) [11]. Furthermore, all these factors are inherently related to the most recent clonal origin interpretation [12], which indicates two main time frames for the development of rHCC: early recurrence, meaning any recurrence within 2 years of resection, which has been associated with tumour factors; and late recurrence, meaning any recurrence after 2 years of resection, which has been associated more with underlying cirrhosis [11]. However, other authors reported different time frames behind the development of rHCC [13,14,15], indicating that more studies on rHCC development should be performed.

While great attention has usually been posted on treatment algorithms for the first HCC [16,17], standardized indications for rHCC are lacking. Thus far, only one proposal from an International Eastern and Western consensus has been published [18]. At the same time, new studies have been recently published on the role of redo surgery in cases of rHCC [19]. These, together with those studies on salvage liver transplantation (SLT) [20,21], appear to be good to take stock of this argument.

## 2. Pattern of Recurrence and Prognostic Significance

After a first curative treatment, HCC has a 5-year recurrence rate of up to 70% [22,23]. The recurrence is related to several different factors. Other than the eventually underlying liver hepatopathy, there are primary tumour characteristics strictly related to aggressiveness and risk of relapse, hesitating in poor survival [24,25]. In terms of pathological appearance, several HCC subtypes have been defined with associated clinical and genetic features [26,27]. The 5th WHO classification defines eight HCC subtypes based on histopathological morphology: steatohepatitic, clear cell, macrotrabecular-massive, scirrhous, chromophobe, fibrolamellar carcinoma, neutrophil-rich, and lymphocyte-rich [27]. On the other hand, six different subclasses (G1–G6) have been distinguished based on gene expression profiles [28]. Combining these characteristics, a particular risk profile could be delineated, as largely detailed, in relation to tumour aggressiveness [24,28]. Among the risk factors, vascular invasiveness (as in the macrotrabecular subtype), TP53 enrichment (G1–G3 subclasses), sarcomatous changes (as in the scirrhous-sarcomatoid subtype) were associated with poor prognosis. On the other hand, chromosomal stability (G4–G6 subtypes) and lymphocytic over neutrophil infiltration (as steatohepatitic) were associated with good prognosis [24,26].

Moreover, recurrence may be characterised by different patterns of relapse considering either the time to recurrence (early versus late) or the type of treatment received (PLR versus PLT). HCC may relapse within the liver (intrahepatic recurrence, IHR) or in extrahepatic sites (EHR). 

IHR is more often the major challenge after a PLR; notably, it is associated with the number and size of HCC nodules as well as with the presence of micro- and macro-vascular invasion [29]. Conversely, EHR mainly consists of pulmonary nodules that, while not being as frequent, are associated with a poorer prognosis than IHR [30]. The latter type of recurrence is more associated with PLT rather than with PLR, particularly when PLT is performed beyond the Milan criteria [31] or in the presence of macrovascular-invasive HCC. HR and EHR may coexist. 

In the last decade, increasing attention has been drawn to IHR patterns [32,33,34], as two different events have been described with distinct prognostic significances [14,35,36]. The first is the event of intrahepatic metastasis (IM) while the second is the event of multicentric occurrence (MO). Sakon et al. [37,38] further categorised IM events as follows: (1) local IM (Figure 1A), where the rHCC develops along the tumour blood flow (TBF) or venous drainage, and (2) systemic IM (Figure 1B), which considers the recurrence caused by circulating tumour cells (CTCs) in the rehoming of the remnant liver. Notably, this IM categorisation was clearly demonstrated in a large European series, where local IM was related to the presence of positive surgical margins after a first resection, while intra-hepatic distant relapse (systemic IM) was related to daughter nodules and microvascular invasion [39].

In contrast, MO appears to be more likely associated with de novo tumour formation due to its tardive recurrence [40]. In addition, MO usually consists of multiple nodular disease with well-differentiated carcinoma surrounding a less-differentiated mass (known as “nodule in nodule” form) and rare vascular involvement (Figure 2) [11,41,42].

Consistently, in terms of relapse timing, early recurrence has been associated with the IM event, while late recurrence is associated with the MO event [11,13,15]. As is already known, in 2003, Imamura et al. [11] identified two peaks of recurrence after PLR: the earlier at 1 year (essentially following an IM mechanism) and the later at 4 years (categorised as MO aetiopathogenesis). Furthermore, three prognostic factors were associated with the early recurrence: non-anatomical resection, microscopic vascular invasion, and elevated serum alpha-fetoprotein (AFP) level > 32 ng/mL. On the other hand, a higher grade of hepatitis activity and well-differentiated multiple tumours were identified as factors related to the late recurrence [11]. These data were further confirmed by other groups [43,44], and a recent meta-analysis reported how the MO group had higher overall survival (OS) and disease-free survival (DFS) than the IM group [32]. 

Along with the aforementioned pathological features and recurrence, enabling the discrimination between IM and MO, great efforts have been spent on identifying molecular changes, such as clonal origin and genetic alterations, to better differentiate these two patterns [12,45]. In recent years, cutting-edge techniques such as next-generation sequencing, allowed for intertumoural genetic heterogeneity to be better defined, demonstrating that IM shared analogous molecular changes with the primary tumour, while MO displayed different genetic mutations [46]. In this scenario, Furuta et al. [47] proposed that a common mutation rate of less than 5% in HCC nodules indicated the MO pattern, while the IM was characterised by more than 5% of shared mutations. These results highlighted that the association of clinical parameters with tumour molecular analysis could improve the accuracy in the diagnosis and management of IM/MO. 

Of note, no guidelines currently include histopathological, genetic and IM/MO patterns as part of the treatment algorithms for rHCC. This lack of guidelines highlights the need for future efforts in individualising postoperative surveillance and postoperative therapies.

## 3. Adjuvant Postoperative Treatments 

Some studies have been conducted to test the role of adjuvant postoperative systemic treatments after PLR or PLT: as HCC is a chemo-resistant tumour, cytotoxic therapies failed to provide any survival benefit and have been abandoned [48]. Similarly, the multikinase inhibitor sorafenib was tested after PLR or ablation in the STORM study, which was a randomised controlled trial, but no effective advantages were recorded [49]. The more recently introduced Lenvatinib has not yet been tested in the adjuvant setting after PLR, with the only study reporting no benefits in terms of rHCC after PLT [50].

A different story might be associated with immune checkpoints inhibitors (ICIs) which are emerging as a treatment option for HCC. There might be a strong rationale for the application of ICIs to prevent rHCC after PLR or ablation since both treatments are known to increase immunogenicity [51]. PLR and ablation may cause the release of tumour-associated antigens and antigen-presenting cells that can activate the cytotoxic effects mediated by CD8+ T cells. This mechanism can be effective in those microscopic daughter nodules left behind a PLR. Such a scenario was reported by Duffy et al. [52], who showed the infiltration of CD8+ cells in untreated HCC nodules after ablation or trans-arterial therapies.

With this rationale, there are different ongoing studies on ICIs in the adjuvant setting after PLR, by using single agents, a combination of two or three agents, or a combination of immunotherapy and an anti-VEGF factor such as bevacizumab. However, some preliminary results are disappointing [53], mainly due to the evidence of the strong immunosuppressive tumour microenvironment of rHCC being associated with immunotherapy resistance [54,55].

Other than medical therapies, it deserves to be mentioned the emerging role of transarterial chemoembolization (TACE) in the adjuvant setting. Even with initial controversial results [56,57], TACE after curative PLR resulted in longer DFS and OS for selected patients with microvascular invasion, as reported from a recent meta-analysis [58]. Specifically, these results were confirmed when a subanalysis was performed in patients with microvascular invasion, multinodular disease and the largest tumour size greater than 5 cm, all risk factors associated with early recurrence [58].

At the moment, neither medical therapies nor interventional radiology techniques have been standardized or commonly recognized to prevent HCC recurrence after PLR. At the least, we must wait for the results of the ongoing clinical trials. 

## 4. Surgical Treatments for rHCC

### 4.1. Redo Hepatectomy

In the case of HCC relapse, most physicians consider recurrence as failure of the curative treatment, hence addressing the patients to palliative care. However, increasing evidence underlines the chance to cure the recurrence, managing it as a first occurrence and resulting in the prolongation of OS [19]. Redo hepatectomy (RH) has been demonstrated as a valid option, as firstly described by Nagasue et al [59]. Due to the improvements in surgical techniques and perioperative care, Chan et al. [60] reported 90-day mortality to be a rare event, with a median rate of 0% (range 0–6%), indicating that the safety profile of RH is equivalent to that of PLR. The key for obtaining the most successful results is to select patients who may benefit more from a second surgical treatment. In this case, there are no clear guidelines, with few groups providing only general recommendations [61]. For this reason, most clinicians are led to restage the recurrence as a first occurrence [62] and several different factors should be considered in this management process [44], such as the number of recurrent nodules, nodule size, IHR or EHR, and gross vascular invasion. On the other hand, there are also factors related to the patient such as: age [18], the functional liver reserve estimable with different methods including the Child–Pugh–Turcotte score [63,64], the BILCHE score [65] or the indocyanine-green retention test [66], platelet count, the AFP-value at the time of recurrence [67] and the value of hepatic venous pressure gradient [68]. Balancing all of these factors, while only about 20% of all rHCC cases were surgically treatable, they were treated with minor hepatectomies in almost 99% of cases, which are interventions that are associated with a lower risk profile [69]. This was previously reported by Chan et al. [60], who finally proposed considering simplicity in the decision process leading to RH, with only two stronger predictors of poor survival outcomes: the presence of vascular involvement and the residual hepatic reserve.

The so-called “test of time”, which may have importance for the risk of developing a second recurrence after RH, is no less important. In this sense, some authors indicate how an early relapse should be regarded as an indirect expression that correlates with a worse outcome [11,13,19,70,71]. Nevertheless, DFS was the only factor found to be independently associated with survival in the systematic review as reported by Chan et al. [60].

Applying these aforementioned selection criteria, RH showed a median OS varying between 22.0 and 71.7 months and a median DFS ranging between 7.0 and 57.0 months as shown in Table 1 [14,70,72,73,74,75,76,77,78,79,80,81,82,83,84,85,86,87,88,89,90,91,92]. 

More recent experiences reported median OS after the resection of rHCC of 56 months, and about half of the patients resected alive at 5 years after surgery [44]. Of note, these results are surely an improvement compared to systemic non-curative treatments, which are addressed in a non-selected population [93].

In terms of surgical technique, if redo open liver resection (ROLR) is still regarded as the standard procedure [73,78], redo laparoscopic liver resection (RLLR) has been gaining increasing attention [94,95]. From a systematic review of the literature [96], RLLR has been associated with outcomes comparable to the first LLR [97] and adequate oncological outcome, resulting in no cases of positive surgical margins in the studies analysed [98,99,100,101,102,103] with results to be confirmed by larger sample sizes.

Whenever both ROLR and RLLR are feasible, it is worth noting the removal of tumour-bearing portal territories, that is at the base of all kinds of anatomical resections, reduces the risk of recurrence intended as local IM [104,105,106]. On the other hand, tumour-vessel detachment (R1vasc surgery), which is not in contradiction with the anatomical conduct of the resection, has been proven to be oncologically suitable and may be useful in the event of rHCC [107,108].

Looking at EHR, especially for lung metastases, surgical resection could be indicated as well, but it is reported to be advantageous only when the recurrence is isolated and the patient has had a DFS > 1 year from the initial LR [109].

There is also the role of ablation techniques for liver-only recurrence cases whenever small nodules (<2 cm) or less than three nodules (the largest <3 cm) occur. In these cases, ablation may be regarded as a curative option following the same indications of a first occurrence [16,110]. However, in a large multicentric recent study, radiofrequency ablation resulted in shorter DFS compared to RH, although no differences in OS were reported [111].

### 4.2. Salvage Liver Transplantation

The standard criteria for liver transplantation in the case of HCC were introduced by Mazzaferro et al. in 1996 [31]. For patients meeting these criteria, the 5-year survival rate was up to 70% with a recurrence rate of less than 15% [112]. About 70% of recurrences are diagnosed within the 2 years following PLT [113], with the lung being the most frequent metastatic site [114,115].

Due to organ shortage, not only the recurrence but the primary HCC is still difficult to approach by PLT in a timely manner. It was from this gap that the work of Majno [20] took place, who estimated that 30% of primary HCC would outgrow the Milan criteria in each 6-month interval (about 5% per month). Following this, they firstly proposed PLR followed by liver transplantation in the case of tumour recurrence or deteriorating liver function, the so-called SLT [20]. After that, the definition of SLT gained several different meanings. SLT could be performed after rHCC following primary resection (as described by Majno) with conceptual difference depending on whether the first HCC was transplantable or not (the so-called “downstaging primary LR”). SLT may also follow PLR before tumour recurrence (“de principe” LT), and the “bridge LT” when PLR is the chosen neoadjuvant treatment before LT [116,117]; however, this attitude had a published experience as a counterpart, showing higher morbidity and mortality of LT after PLR [118]. Additionally, SLT is sometimes performed because of irreversible liver failure after resection (“rescue” LT). Finally, the possibility of performing an SLT following a PLT, when at recurrence, rHCC is still within transplantable criteria, has been raised [119].

Theoretically, rHCC may have a worse biological behaviour; it is still also controversial whether the same transplantation criteria should be applied as for PLT. For this reason, generally approved criteria for SLT are missing, and several efforts have been made looking for the most suitable patient. Accordingly, Zhang et al. [120] compared various existing criteria such as the Milan criteria, the University of California San Francisco criteria (UCSF) [121] and the model for end-stage liver disease (MELD) [122], concluding that the Milan criteria are those associated with the best outcome in the setting of SLT. On the contrary, de Haas et al. [123] proposed a patient with a higher MELD score and no preoperative bridge therapy, such as trans-arterial chemo-emboliation, no postoperative complication after a PLR and low T-stage in the primary resected HCC as the best candidate for SLT. While no agreement exists on the selection criteria for SLT, it is well accepted that the number and the size of rHCC are important factors to be considered during decision making.

Whether a living donor liver transplant (LDLT) rather than a deceased donor liver transplant (DDLT) should be preferred for SLT is still under debate [124]. While no large experiences have been described and no randomized trials have been conducted, specific pros and cons of LDLT and DDLT tested in the case of PLT may be shifted in the direction of SLT [125,126]. In particular, the oncological benefit of LDLT has been argued, since it has been associated with an increased recurrence rate, probably associated with the lack of biological selection made by the test of time [127]. These results are at least controversial, particularly if data reported by a recent multicentric study are considered [128]. In this multicentric intention-to-treat analysis, living donor for PLT resulted to be an independent protective factor for overall death.

If large efforts have been made in analysing recurrence risk after PLT [129,130,131], the lack of data in terms of recurrence after SLT makes comparison challenging. Looking at the PLT experience, significant predictors of recurrence appeared to be the following: poor tumour grading, microvascular invasion and the diameter of the largest tumour [132], with the most favourable prognosis reached in the case of unifocal liver recurrence with concomitant low AFP level [133].

Conflicting results have been reported by the role of PLR preceding transplant (downstaging PLR) since it has been argued that it may represent a possible risk factor for recurrence itself, and for this reason, it should be reserved to limited cases [118,132,134]. However, if bridge liver resection can lead to tumour manipulation, and to longer and more bleeding transplant surgery, it also gives the possibility, throughout the pathological examination of the surgical specimen, to select patients on the basis of the aforementioned risk factors [118,132,134].

Apart from PLR, the role of other bridging therapies has been reported as well. Even if their potential advantages have been analysed mainly in relation to patients awaiting for PLT, such benefits can be translated to patients on the waiting list for a SLT, since no specific guidelines are present. These should be considered as a sort of neoadjuvant treatment, mainly consisting of several types of locoregional procedures (TACE, transarterial radioembolization, stereotactic body radiotherapy), to avoid tumour progression, thus reducing the drop-out rate of patients on the waitlist. For PLT, bridge therapies have been proposed when the expected waiting time on the list exceeds 6 months with a complete response to LRT, resulting in a significant reduction of dropout at 3, 6 and 12 months [135]. On the other hand, as proposed for PLT, an unsatisfactory response to these treatments may represent a criterion for prioritizing patients on the waitlist [136].

### 4.3. RH vs. SLT: Which Is the Best Option?

Several studies have reported a comparison between SLT and RH in the case of rHCC (Table 2) [119,137,138,139,140]. Clearly, there are many factors to be considered, not least the experience of a given centre in performing both liver resection and transplantation. Apart from the specific centre criteria, the length of the waiting list, the general organ shortage, and the type of LT (LDLT versus DDLT), the patient’s age remains an important factor to be taken into consideration. While there is no longer a given age limit for LT in some liver transplant centres, being old may still represent per se a contraindication for LT and for SLT. For these reasons, RH appears to be more feasible than SLT, in particular for older patients. Moreover, RH could be performed when some prohibitive conditions for liver transplantation (e.g., macrovascular invasion) are present, along with the concept of the “therapeutic hierarchy” as historically endorsed by the Asia-Pacific treatment algorithm [61]. On the other hand, SLT may be considered if unresectable cases present with conditions that still fall in specific centre criteria of transplantability. 

More in detail, when looking at the survival benefit of RH versus SLT, some discrepancies from the review of the literature emerged. In 2020, Fang et al. [138] described their experience with 124 patients undergoing RH or SLT after rHCC following PLR, reporting recurrence-free survival and OS rates that were significantly higher in the SLT group. This was despite the two groups differing in terms of preoperative total bilirubin levels, number of multiple tumours (higher in the SLT group) and HCC size (higher in the RH group). However, the advantage of SLT over RH was not confirmed when only patients with higher AFP at recurrence (>100 ng/mL) were considered, indicating that the selection process for RH versus SLT is complex. Furthermore, Fang et al. [139] reported longer operation time, increased blood loss and blood replacement, prolonged in-hospital stays and postoperative morbidity in the case of SLT; in addition, perioperative mortality was not reported for the groups.

In another HCC patient cohort previously treated with PLR or ablation, Ma et al. [138] reported, with the use of propensity score matching (PSM), that the 5-year OS and tumour-free survival were higher in the SLT group (with no distinction between DDLT and LDLT) versus RH. A similar propensity score matching analysis was also performed by Yoon et al. [140], who focused on a possible benefit of LDLT compared to RH. This resulted in LDLT having a longer DFS than RH even after a PSM. Along this line, some other papers deserve to be mentioned. Wang et al. [40] analysed 840 patients and compared SLT with curative loco-regional therapy (CLRT). The SLT was found to be associated with significantly higher 3- and 5-year DFS than in the CRLT group, but no OS benefit was achieved. SLT resulted in greater intraoperative blood loss and longer in-hospital stays [40]. Kostakis et al. [141] reported a similar analysis on 516 patients. They demonstrated that SLT was burdened by higher rates of postoperative complications and mortality, even if the higher rate of death did not reach statistical significance. Yamashita et al. [137] retrospectively studied the efficacy of SLT (exclusively LDLT) versus RH in a population of 146 patients with rHCC treated by PLR. Once again, no statistically significant differences in OS were found between the two treatments with the SLT showing a longer 5-year DFS rate (86% versus 16%). Lim et al. [119] conducted an intention-to-treat analysis in which the patients collected were previously transplanted due to resectable and transplantable primary and recurrent HCCs. The 90-day mortality was 4% for the SLT group and 0% in the RH (*p* = 0.007). The 5-year DFS rates were 72% for the SLT and 18% for the RH (*p* < 0.001). However, at a 5-year intention-to-treat analysis of OS, no statistically significant differences were recorded between SLT and RH [119].

From all these reported experiences, there is need for a clear consensus that can precisely define the proper treatment strategy in cases of rHCC, as performed for primary occurrence disease. The multispecialistic armamentarium from medical, radiological and surgical options should be precisely addressed in a tailored manner on patient characteristics. This highlights the demanding role of a multidisciplinary approach with the multimodality care representing the best solution in the high variability of recurrence presentation. The growing role of histopathologic and genetic prognostic characteristics allows surgical treatment to no longer be a technical problem, and this should be widely discussed in a case-by-case manner. The test of time may represent an underestimated concept, with the IM/MO characterization of recurrence still misunderstood. From the evidence reported and from the evidence coming from the ongoing trials, particularly on immunotherapies strategies, a new important consensus treatment algorithm should be stated.

## 5. Conclusions

In conclusion, according to our critical review of the literature, it is clear that both RH and SLT are feasible procedures in the setting of rHCC. If SLT consists, as expected, in a more complex procedure burdened by increased postoperative morbidities and mortality, it is also associated with longer DFS. Conversely, RH represents a valid chance of cure, especially for the oldest patients or patients with comorbidities with limited postoperative risks. While waiting for the new and promising immunotherapies in the adjuvant setting, further randomized clinical trials comparing SLT versus RH should be performed in the setting of rHCC.

## Figures and Tables

**Figure 1 cancers-15-00508-f001:**
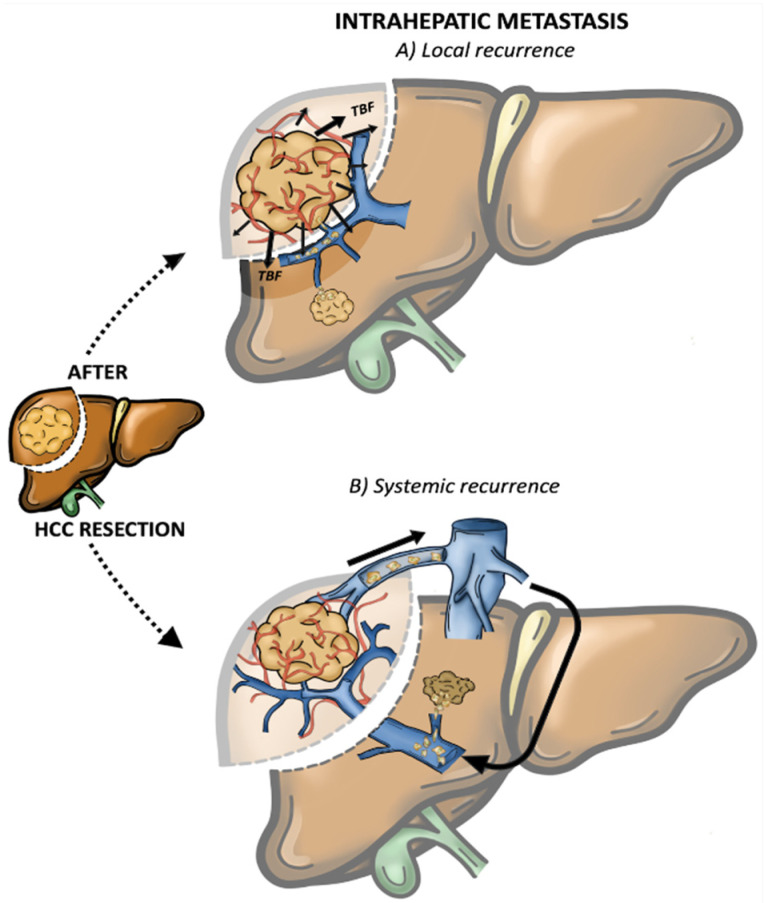
Intrahepatic metastasis (IM). (**A**) *Local recurrence:* the rHCC develops along the tumour blood flow (TBF) (**B**) *Systemic recurrence:* caused by circulating tumour cells (CTCs) in the rehoming of the remnant liver.

**Figure 2 cancers-15-00508-f002:**
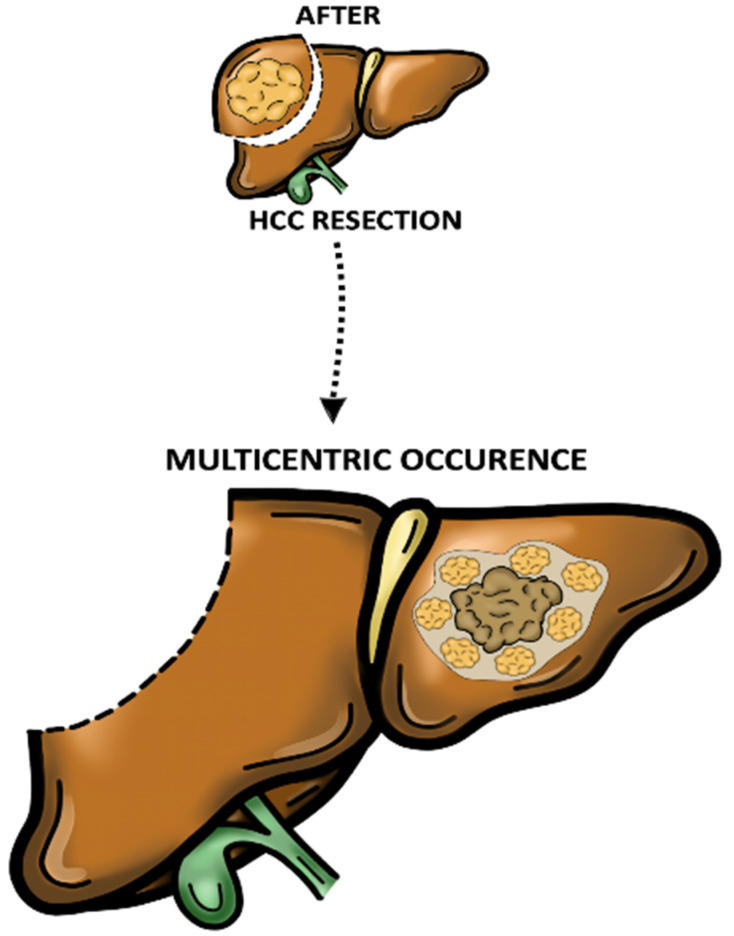
Multicentric occurrence (MO). Multiple nodular disease with well-differentiated carcinoma surrounding a less-differentiated mass (known as “nodule in nodule” form).

**Table 1 cancers-15-00508-t001:** Published papers on redo hepatectomy for recurrent hepatocellular carcinoma (after liver resection at first occurrence) in the last 20 years.

First Author	N° of Cases	Extent of Resection	Survivals					
		Minor Resection n (%)	Major Resection n (%)	Median DFS	Median OS	1-Year Survival (%)	3-Year Survival (%)	5-Year Survival (%)
Matsuda ● (2001) [79]	25	NR	NR	NR	NR	84	68	56
Nakajima (2001) [80]	12	11 (92)	1 (8)	9	NR	90	80	55
Sugimachi (2001) [81]	78	NR	NR	NR	NR	NR	83	48
Minagawa (2003) [78]	67	61 (91)	6 (10)	12	NR	93	70	56
Sun (2005) [77]	57	NR	NR	NR	26	70	61	31
Kobayashi (2006) [82]	60	NR	NR	18	56	97	74	53
Itamoto (2007) [83]	84	73 (87)	11 (13)	9	60	88	67	50
Shimada ^α^ (2007) [76]	13	NR	NR	NR	63	97	67	57
Tralhão (2007) [84]	16	13 (81)	3 (19)	27	34	89	46	31
Kubo (2008) [85]	51	NR	NR	15	55	94	77	52
Liang (2008) [75]	44	42 (98)	1 (2)	NR	30	79	45	28
Kawano (2009) [86]	13	NR	NR	NR	36	100	50	26
Nagano (2009) [74]	24	NR	NR	NR	62	92	73	51
Wu (2009) [87]	149	148 (99)	1 (1)	27	Not reached	91	78	56
Tsujita (2010) [88]	121	NR	NR	NR	Not reached	97	88	83
Zhou (2010) [73]	37	33 (89)	4 (11)	NR	50	95	70	44
Faber (2011) [89]	27	25 (89)	3 (11)	17	36	96	70	42
Roayaie (2011) [70]	35	29 (83)	6 (17)	32	Not reached	90	67	67
Umeda (2011) [90]	29	NR	NR	NR	66	93	67	56
Chok (2012) [91]	47	41 (87)	6 (13)	11	54	81	55	44
Ho (2012) [72]	54	NR	NR	NR	Not reached	96	84	72
Huang (2012) [14]	82	NR	NR	7	22	71	41	22
*Chan (2013)* [60]	1125	* 89 * * (81–99) *	* 11 * * (1–19) *	* 15 * * (7–32) *	* 52 * * (22–66) *	* 92 * * (70–100) *	* 69 * * (41–88) *	* 52 * * (22–83) *
**Tabrizian (2015)** [44]	44	44 (100)	0 (0)	NR	56	NR	NR	47
**Famularo (2021)** [19]	156 *	NR	NR	57	Not reached	100	70.3	52.7
**Yoh (2021)** [92]	128°	100 (90)	11 (10)	NR	71.7	91	66.9	55.1
Median value ^♦^ (range)	44 (12–156)	89.5 (81–100)	10.5 (0–19)	16 (7–57)	54 (22–71.7)	92 (70–100)	69 (41–88)	52 (22–83)

NR = not recorded, DFS = disease free survival, OS = overall survival; ● includes 7 microwave ablations; ^α^ includes 1 patient with extrahepatic recurrence; * 79 of 156 patients underwent ablation, 17 of 156 had concomitant extrahepatic recurrence, ° 55 of 128 had concomitant extrahepatic recurrence, and 17 of 128 had extrahepatic recurrence only. For studies in **BOLD**, OS are intended as survival after recurrence (SAR); in *italics:* median (IQR) results from a systematic review (only the total of available data from studies analysed was reported); ^♦^ Calculated median value and ranges of all reported values with the exception of Chan (2013) [60].

**Table 2 cancers-15-00508-t002:** Studies comparing redo hepatectomy versus salvage liver transplantation.

First Author	N° of Cases	Treatment at Recurrence	Survivals					
		RH	SLT	RFS/DFS (RH)	RFS/DFS (SLT)	OS (RH)	OS (SLT)	*p* Value (RFS/DFS, OS)
Yamashita (2015) [137]	159	146	13	16% (5 y)	81% (5 y)	61% (5 y)	75% (5 y)	0.0002, 0.1714
Lim (2017) [119]	99	81	18	18% (5 y)	72% (5 y)	71% (5 y)	71% (5 y)	<0.001, 0.99
Ma (2018) ^α^ [138]	144	108	36	32.8% (5 y)	71.6% (5 y)	48.3% (5 y)	72.8% (5 y)	<0.001, 0.01
Fang (2020) [139]	124	78	46	16.0 (8.0–27.3)	32.0 (12.8–45.0)	23.0 (15.0–32.5)	36.5 (20.3–45)	<0.01, <0.01
Yoon (2021) ^α^ [140]	84	42	42	27.9 (5 y)	78% (5 y)	62.2% (5 y)	89.2 (5 y)	<0.001, <0.001
**Median value ^♦^ (range)**	124 (84–159)	81 (42–146)	36 (13–46)	22.95 (16–32.8)	75 (71.6–81)	61.6 (48.3–71)	73.9 (71–89.2)	

RH = Redo-Hepatectomy, SLT = Salvage Liver Transplantation, RFS = Recurrence-Free Survival, DFS = disease-free survival, OS = Overall Survival; data are given as percentage (%) or median (range) unless otherwise noted. ^α^ Population after propensity score matching; ^♦^ Calculated median value and range of all reported values with the exception of “survivals” of Fang (2020) [139].

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
