# Peer review of "Surgical Strategies for Recurrent Hepatocellular Carcinoma after Resection: A Review of Current Evidence"

_cancers, 2023, doi:10.3390/cancers15020508_

Round 1

Reviewer 1 Report

First of all I will personally congratulate the authors to embark on such impart topic. As a transplant and hepatobiliray surgeon, i have seen patient with HCC almost daily in my practice, and majority of them (80%) deem not to be candidate for resection due to liver cirrhosis, "end stage liver disease" severe portal hypertension, splenomegaly, thrombocytopenia OR multiple other morbidities, frailty, ageing ect.. with the only option being liver transplant.

The other 20% whom undergoing surgical resection, there recurrence rate has been very high in 1,3, and 5 years 15-30- 70% respectively. 

but few patients undergo surgical re resection 

unfortunately, due to shortage of organ, organ allocation, most of them will succumb with the disease and not make it.

until recently with immunotherapy approach there is a paradigm shift for the treatment algorithm for HCC and open new opportunity for these patient population. 

Great job and thank you for comprehensive approach to tackle this complex disease.

If you add elaborate about different type and variants of HCC in your paper this, make it very conclusive.

Author Response

First of all I will personally congratulate the authors to embark on such impart topic. As a transplant and hepatobiliray surgeon, i have seen patient with HCC almost daily in my practice, and majority of them (80%) deem not to be candidate for resection due to liver cirrhosis, "end stage liver disease" severe portal hypertension, splenomegaly, thrombocytopenia OR multiple other morbidities, frailty, ageing ect.. with the only option being liver transplant.The other 20% whom undergoing surgical resection, there recurrence rate has been very high in 1,3, and 5 years 15-30- 70% respectively but few patients undergo surgical re resection. Unfortunately, due to shortage of organ, organ allocation, most of them will succumb with the disease and not make it. Until recently with immunotherapy approach there is a paradigm shift for the treatment algorithm for HCC and open new opportunity for these patient population. Great job and thank you for comprehensive approach to tackle this complex disease.

If you add elaborate about different type and variants of HCC in your paper this, make it very conclusive.

Reply. We would thank the reviewer for the positive comments. We improved our paper following the suggestion and adding a paragraph about different variants of HCC, regarding their recurrence patterns. The changes were highlighted in red.

Reviewer 2 Report

Milana et al. performed a nice review on the current evidence on surgery for recurrent HCC after resection. The review is comprehensive enough, however it would be nice to have a larger discussion of the evidence as well as a summary at the end, maybe including IM/MO patterns.

My comments:

-          Simple summary

o   “both represent” not “represent both”

o   (HCC) not (HCC9

o   Instead of “resume”, “summarize”?

-          Introduction

o   The etiology of HCC seems to be based on old data and not necessarily reflective of the changing etiological landscape

-          Pattern of recurrence

o   Figure 1 and 2 are not mentioned in the text

o   “Of note, currently no guidelines include IM/MO patterns as part of the treatment al-gorithms for rHCC. This lack highlights the need for future efforts in individualizing post-operative surveillance and postoperative therapies.” As previously mentioned, it would be interesting to tie this into the following discussion on LR/LT

-          Adjuvant postoperative treatments

o   “The more recently introduced Lenvatinib has not been yet tested in the adjuvant setting after PLR” > Zhou et al presented a preliminary analysis of adjuvant lenvatinib at ASCO 2022 DOI: 10.1200/JCO.2022.40.16_suppl.e16158 Journal of Clinical Oncology 40, no. 16_suppl (June 01, 2022) e16158-e16158. , however you are correct that no study has been formally published

o   For the sake of completeness, a mention of adjuvant TACE/IR strategies could be added

o   Ref 45, is it correct? The cited article does not seem to mention results from a trial

o   “it is unlikely that they will be available soon” > this could be the case, but there are several ongoing trials whose results will soon be available

-          Surgical treatments for rHCC

o   Redo-hepatectomy

§  Instead of “addressing the patients”, “referring the patients”?

§  “Common sense leads most clinicians to re-stage the recurrence as a first occurrence” unclear sentence, would rephrase

§  “Balancing all these factors, if only about 20% of all rHCC resulted surgically treatable, they were in almost 99% of cases treated with minor hepatectomies meaning with interventions associated with a lower risk profile” ref?

§  “a given early relapse” unclear

§  “Yet, DFS was the only factor found to be independently associated with survival in the systematic review as reported by Chan et al.” is this supposed to be in contrast or in agreement with the previous sentence?

§  “Applying these aforementioned selection criteria, RH showed a median OS varying between 22,0-71,7 months and median DFS ranging between 7,0-57,0 months as shown in table 1”: what selection criteria are you referring to? These ranges are quite wide, difficult to make any significant conclusion

§  How were the studies in Table 1 selected? How was the median value calculated?

§  LLR84 > LLR [84]?

§  “To be mentioned that anatomical resection, whenever both ROLR or RLLR are feasi-ble, reduces the risk of IM [91,92]. On the other hand, tumor-vessel detachment (R1vasc surgery), which is not in contradiction with the anatomical conduct of the resection, has been proven as oncologically suitable then possibly useful in the event of rHCC [93,94].” > unclear paragraph. Does anatomical resection reduce the risk of IM after primary LR or after recurrence?

§  Ablation techniques could be mentioned for completeness

o   Salvage liver transplantation

§  PLT97 > PLT [97]?

§  The higher recurrence after LDLT has recently been challenged doi:10.1001/jamasurg.2021.3112

§  “Otherwise, conflicting results have been reported by the role of PLR preceding the transplant (downstaging PLR), as a possible risk factor for recurrence [115,116]” while a bit outside of the scope of this review, it might be interesting to briefly mention the mechanistic link between PLR and recurrence

§  “However, the most favorable prognosis is reached in case of unifocal liver recurrence or limited ex-trahepatic one (lung, parietal) with concomitant low AFP level” > I could not find this information in the provided reference. I’m not sure this paragraph is in the right spot: it makes little sense to discuss extrahepatic recurrence in the section on SLT, as extrahepatic disease would be a contraindication to SLT.

§  It might be valuable to add a paragraph on the available evidence on bridge therapies to SLT

o   RH vs SLT

§  It should be mentioned that RH and SLT are not always applicable to the same patient population. For example, RH requires for the tumor to be resectable and for the patient to have sufficient liver function but, differently from LT, can be performed in select patients with concomitant extrahepatic recurrence (and, as you mentioned, older patients and patients unfit fot LT). Selection criteria for LT are stricter, but LT can be considered in patients with liver function not compatible with resection.

§  “While nowadays in some liver transplant centers there is not anymore a given age limit for LT, it is clear that the transplantability is inversely related to the patient’s age. Without men-tioning the frequent association of advanced age and comorbidities, being old may still represent per se a contraindication for LT, and then also for SLT.” > not sure this sentence is completely accurate. What does “the transplantability is inversely related to the patient’s age” mean? There is certainly an age limit after which transplantation becomes futile (i.e. natural life expectancy is lower than the minimum survival benefit required for LT), however many other factors are involved in the selection of LT candidates and this sentence seems reductive.

§  It should be mentioned that a reason to strongly consider SLT after LR is to treat the underlying liver disease that will likely continue to be a source of carcinogenesis if not addressed, and may eventually lead to liver decompensation and liver failure. This is also linked to your consideration on age: since LT has a theoretical age limit, each patient with HCC should be considered for LT, especially at recurrence, as that may be the patient’s last chance for a definitive curative treatment of both their HCC and their liver disease.

§   Table 2 is not mentioned in the text; how was the median calculated? “Population after propensity-score matching;” > does this refer to Ma and Yoon?

§  “When looking at the survival benefit of RH versus SLT, some difficulties emerge” difficulties?

Author Response

Milana et al. performed a nice review on the current evidence on surgery for recurrent HCC after resection. The review is comprehensive enough, however it would be nice to have a larger discussion of the evidence as well as a summary at the end, maybe including IM/MO patterns.

We are thankful with the reviewer for the detailed suggestions. We followed the indications given to improve the paper and refine our work as explicated in the following tables, where the comments have been replied point by point. The discussion was valorized as well, including the matter of missing guidelines that could comprehend the concept of IM/MO recurrence pattern. All the changes were highlighted in red.

My comments:

-          Simple summary: it was broaden as point out by the editorial office, following the reviewers’ comments.

“both represent” not “represent both”

It has been corrected as suggested

o   (HCC) not (HCC9

Correct spell checked

Instead of “resume”, “summarize”?

Modified as suggested

-          Introduction

o   The etiology of HCC seems to be based on old data and not necessarily reflective of the changing etiological landscape

The changing of the etiological landscape has been detailed in the introduction

-          Pattern of recurrence

o   Figure 1 and 2 are not mentioned in the text

The figures were already mentioned in the text. We modified the style in bold type to make the mention clearer

“Of note, currently no guidelines include IM/MO patterns as part of the treatment al-gorithms for rHCC. This lack highlights the need for future efforts in individualizing post-operative surveillance and postoperative therapies.” As previously mentioned, it would be interesting to tie this into the following discussion on LR/LT

We agree with the reviewer that it would be worth to stress this topic. We analyzed it in detail in the discussion.

-          Adjuvant postoperative treatments

“The more recently introduced Lenvatinib has not been yet tested in the adjuvant setting after PLR” > Zhou et al presented a preliminary analysis of adjuvant lenvatinib at ASCO 2022 DOI: 10.1200/JCO.2022.40.16_suppl.e16158 Journal of Clinical Oncology 40, no. 16_suppl (June 01, 2022) e16158-e16158. , however you are correct that no study has been formally published

We thank the reviewer for the comment. We appreciate that, hoping results will be published soon.

Fo For the sake of completeness, a mention of adjuvant TACE/IR strategies could be added

We followed the suggestion adding a paragraph about TACE in the adjuvant setting

 Ref 45, is it correct? The cited article does not seem to mention results from a trial

The reference had been appropriately modified.

 “i”it is unlikely that they will be available soon” > this could be the case, but there are several ongoing trials whose results will soon be available

We reformulated the phrase

 Surgical treatments for rHCC

Redo-hepatectomy

·       Instead of “addressing the patients”, “referring the patients”?

It was modified as suggested

“C”Common sense leads most clinicians to re-stage the recurrence as a first occurrence” unclear sentence, would rephrase

The expression was rephrased

§  “Balancing all these factors, if only about 20% of all rHCC resulted surgically treatable, they were in almost 99% of cases treated with minor hepatectomies meaning with interventions associated with a lower risk profile” ref?

The appropriate reference was added

§  “a given early relapse” unclear

It has been reformulated

§  “Yet, DFS was the only factor found to be independently associated with survival in the systematic review as reported by Chan et al.” is this supposed to be in contrast or in agreement with the previous sentence?

An extended english revision was done. In particular, this was rephrased.

§  “Applying these aforementioned selection criteria, RH showed a median OS varying between 22,0-71,7 months and median DFS ranging between 7,0-57,0 months as shown in table 1”: what selection criteria are you referring to? These ranges are quite wide, difficult to make any significant conclusion

As stated in the text, there are no clear guidelines for rHCC. For this reason, in terms of “selection criteria”, we referred to the general recommendations and the criteria employed for the staging of HCC first occurrence, mentioned just before in the same paragraph.

Data reported come from a review of the literature and are not intended to be used to make any conclusions. The authors would just summarize the state of art about the topic covered by the paper.

§  “How were the studies in Table 1 selected? How was the median value calculated?

Table 1 reports the studies on redo hepatectomy of recurrent hepatocellular carcinoma. It is based on a review of the literature published in the last 20 years in English. The median is simply the median value with IQR of the data reported by the different authors. Please note that these details have been now included in the Table notes.

§    LLR84 > LLR [84]?

Parenthesis added

§    “To be mentioned that anatomical resection, whenever both ROLR or RLLR are feasi-ble, reduces the risk of IM [91,92]. On the other hand, tumor-vessel detachment (R1vasc surgery), which is not in contradiction with the anatomical conduct of the resection, has been proven as oncologically suitable then possibly useful in the event of rHCC [93,94].” > unclear paragraph. Does anatomical resection reduce the risk of IM after primary LR or after recurrence?

To better explain the concept, the phrase was re-elaborated, and a citation was added.

Ablation techniques could be mentioned for completeness

A paragraph about ablation techniques for recurrent HCC was added

Salvage liver transplantation

§   PLT97 > PLT [97]?

Parenthesis added

§   The higher recurrence after LDLT has recently been challenged doi:10.1001/jamasurg.2021.3112

We thank the reviewer for suggesting us this recent article. We appropriately reformulated the paragraph.

§   “Otherwise, conflicting results have been reported by the role of PLR preceding the transplant (downstaging PLR), as a possible risk factor for recurrence [115,116]” while a bit outside of the scope of this review, it might be interesting to briefly mention the mechanistic link between PLR and recurrence

We perfectly agree that this concept is far from the aim of the review. However few words had been spent on the topic.

§   “However, the most favorable prognosis is reached in case of unifocal liver recurrence or limited ex-trahepatic one (lung, parietal) with concomitant low AFP level” > I could not find this information in the provided reference. I’m not sure this paragraph is in the right spot: it makes little sense to discuss extrahepatic recurrence in the section on SLT, as extrahepatic disease would be a contraindication to SLT.

The phrase was reformulated and moved just above.

§    It might be valuable to add a paragraph on the available evidence on bridge therapies to SLT

A specific paragraph was added.

RH vs SLT

 §  It should be mentioned that RH and SLT are not always applicable to the same patient population. For example, RH requires for the tumor to be resectable and for the patient to have sufficient liver function but, differently from LT, can be performed in select patients with concomitant extrahepatic recurrence (and, as you mentioned, older patients and patients unfit fot LT). Selection criteria for LT are stricter, but LT can be considered in patients with liver function not compatible with resection.

We agree with such point of view, and we detailed in the text. However, the most recent guidelines do not consider resection for patients with concomitant extrahepatic disease, believed to be poor candidates and at least suitable for TACE. Anyway, RH could represent a chance for patients with some other conditions, as macrovascular invasion, that represent absolute contraindication for LT. This point has been underlined in the text.

 §  “While nowadays in some liver transplant centers there is not anymore a given age limit for LT, it is clear that the transplantability is inversely related to the patient’s age. Without men-tioning the frequent association of advanced age and comorbidities, being old may still represent per se a contraindication for LT, and then also for SLT.” > not sure this sentence is completely accurate. What does “the transplantability is inversely related to the patient’s age” mean? There is certainly an age limit after which transplantation becomes futile (i.e. natural life expectancy is lower than the minimum survival benefit required for LT), however many other factors are involved in the selection of LT candidates and this sentence seems reductive.

Thank you for your observation. The sentences has been now modified to make the reader better understand the point.

 §  It should be mentioned that a reason to strongly consider SLT after LR is to treat the underlying liver disease that will likely continue to be a source of carcinogenesis if not addressed, and may eventually lead to liver decompensation and liver failure. This is also linked to your consideration on age: since LT has a theoretical age limit, each patient with HCC should be considered for LT, especially at recurrence, as that may be the patient’s last chance for a definitive curative treatment of both their HCC and their liver disease.

We firmly agree with the reviewer about this point, but we believe that in this case, speaking about of SLT and RH (a re-treatment) those kinds of consideration may be unnecessary. We are describing concepts about treatment at recurrence and all the considerations about transplant advantage over surgical resection are taken for granted.

§   Table 2 is not mentioned in the text; how was the median calculated? “Population after propensity-score matching;” > does this refer to Ma and Yoon?

The table was already mentioned in the text. We modified the style in bold type to make it clearer. The median is simply the median value with IQR of the data reported by the different authors. Both Ma and Yoon performed propensity score matching in their analyses, this was better clarified.

 §  “When looking at the survival benefit of RH versus SLT, some difficulties emerge” difficulties?

We modified the term with “discrepancies” to underline how the results coming from the review of literature are conflicting.

Ho un po’ allungato la discussione, ma eviterei di rifare una review della letteratura da capo. Penso i concetti ci siano tutti, spero vada bene

Round 2

Reviewer 2 Report

I'm satisfied with the changes made to the content.

I would still recommend a language revision of the text